# Effect of the Electric Field on the Biomineralization of Collagen

**DOI:** 10.3390/polym15143121

**Published:** 2023-07-22

**Authors:** Fiorella Ortiz, Antonio Díaz-Barrios, Zoraya E. Lopez-Cabaña, Gema González

**Affiliations:** 1School of Chemical Sciences and Engineering, Yachay Tech University, Urcuquí 100119, Ecuador; 2Institute of Chemistry of Natural Resources, Universidad de Talca, Talca 3460000, Chile; zlopez@utalca.cl; 3School of Physical Sciences and Nanotechnology, Yachay Tech University, Urcuquí 100119, Ecuador

**Keywords:** biomineralization, collagen, electric field, double-diffusion, hydroxyapatite

## Abstract

Collagen/hydroxyapatite hybrids are promising biomimetic materials that can replace or temporarily substitute bone tissues. The process of biomineralization was carried out through a double diffusion system. The methodological principle consisted in applying an electric field on the incubation medium to promote the opposite migration of ions into collagen membranes to form hydroxyapatite (HA) on the collagen membrane. Two physically separated solutions were used for the incubation medium, one rich in phosphate ions and the other in calcium ions, and their effects were evaluated against the traditional mineralization in Simulated Body Fluid (SBF). Pre-polarization of the organic membranes and the effect of incubation time on the biomineralization process were also assessed by FTIR and Raman spectroscopies.Our results demonstrated that the membrane pre-polarization significantly accelerated the mineralization process on collagen. On the other side, it was found that the application of the electric field influenced the collagen structure and its interactions with the mineral phase. The increment of the mineralization degree enhanced the photoluminescence properties of the collagen/HA materials, while the conductivity and the dielectric constant were reduced. These results might provide a useful approach for future applications in manufacturing biomimetic bone-like materials.

## 1. Introduction

Bone defects such as fractures, tumors, traumas are problems in orthopedics that require clinical treatment and surgical procedures. The use of bone substitution materials is a good option for filling these defects. However, the limited availability of bone substitutes, high operating costs, and immune rejection limit their applications. Tissue engineering has been extensively investigated for the development of synthetic biomimetic materials that can replace or temporarily substitute bone tissues.

The essential characteristic of a bone substitution material must be its ability to simulate the native bone structure and mimic its composition. Collagen/HA hybrid materials are promising candidates for this application with proven biocompatibility, biodegradability, enhanced mechanical properties, and good bone repair potential [1].

However, most of the available fabrication approaches do not allow obtaining mineralized collagen scaffolds with the organizational complexity characteristic of bone, and the deeper diffusion of the mineral precursors inside of the collagen matrix remains challenging [2]. Several approaches have been proposed to overcome these drawbacks. One of the most impressive strategies is double diffusion systems. Iijima et al. [3] studied collagen mineralization obtained from the tendon using a glass bottle filled with Ca(CH_3_COO)_2_ solution and a collagen slice attached to the bottle mouth. Then, the plastic bottle was placed upside down in a solution of NH_4_H_2_PO_4_ + (NH_4_)_2_HPO_4_ (1:1 molar ratio). In this system, mineral ions diffused into the collagen fibers from opposite sides of the disk depending on the ion concentration, medium pH, and the thickness and orientation of the collagen fibers. Besides, Dorvee and co-workers [4] designed and evaluated an optimized double-diffusion system for analyzing mineral crystal growth and nucleation in different hydrogels.

Moreover, some studies have focused on implementing an electrophoresis approach to the double diffusion system to improve the biomineralization process [5]. Watanabe et al. [6] described hydroxyapatite formation in agarose hydrogels after the application of an electric field using a conventional electrophoresis apparatus. Furthermore, Heinemann et al. [7], inspired by Watanabe’s study, developed an electric field-assisted double migration system to study the formation of hydroxyapatite in carboxymethylated gelatin gels. The system consisted of a three reservoirs chamber where the gelatin gel was placed as a barrier between the two reservoirs of calcium and phosphate ions whose opposite diffusion caused the mineral precipitation. The study’s findings suggest that the electric field enhances the mineralization process. Kim et al. [2] found that the use of pulsed electrical stimulation markedly increase the diffusion of ionic components of bodily fluids (charged precursors) through the inner collagen surface for bone mineralization. The results suggest that pulsed electrical signals can accelerate the nucleation of calcium phosphate nanocrystals in/on collagen, allowing for better control of their spatial distribution at the microscale.

On the other hand, the bone itself is a piezoelectric material, which means that mechanical deformation causes an electrical negative and positive polarization in its structure. The hydrogen bonds in HA and collagen are responsible for the polarizability properties of native bone [8]. Marino et al. [9] proved that collagen films obtained from electrodeposition and evaporation exhibits piezoelectric properties similar to bone but to a lower degree. Likewise, Kabir et al. [10] noted that collagen charge polarization is one of the mechanisms by which electric stimulus improves bone healing and enhances biomineralization. The separation between positive and negative charges in the collagen matrix could enhance the electrostatic interactions between charged precursor molecules and the organic phase. Therefore, the electrical polarization of collagen membranes before biomineralization (pre-polarization) by applying an electric field directly on the membranes could positively affect the mineral deposition rate.

In general, numerous studies have explored the potential of electrical stimulation to promote bone healing and remineralization [11,12,13], including in vitro investigations that have used collagen samples. Nevertheless, little is known about the impact of applying an electric field directly to collagen membranes leveraging their piezoelectric properties to promote mineralization. Hence, this study aims to assess the effect of pre-polarization of collagen on biomineralization and the comparison of the electrophoretic process against the conventional method by immersion in simulated body fluid (SBF). The methodological principle consists of a double-diffusion system equipped with an electric field on the solution medium to promote the opposite migration of calcium and phosphate ions into collagen membranes that act as an organic template for mineral deposition. Additionally, the effect of incubation time and the mineralization medium are evaluated.

## 2. Materials and Methods

### 2.1. Materials

The following reagents were purchased from Sigma-Aldrich, Saint Louis, MI, USA. Acetic acid 96% purity, sodium hydroxide analytical grade, NaCl 99.5%, NaHCO_3_ 99.5%, KCl 99.5%, K_2_HPO_4_ 99.0%, MgCl_2_ 98%, CaCl_2_ 95%, NaSO_4_ 99%, and C_4_H_11_O_3_N (Tris) 99.0% were purchased from Sigma-Aldrich. Ethanol analytical grade 99.0%, acetic acid 96.0%, Socium hydroxide analytical grade 98.0% Collagen type 1 was obtained from Sprague-Dawley rat tail tendons.

### 2.2. Purification of Type I Collagen from Rat Tail Tendons

Type I collagen was extracted from rat tail tendon following the modification of a protocol described by González et al. [14]. Briefly, five to six Sprague-Dawley rat tails were washed with absolute ethanol to carefully pulled out the tendons. Subsequently, a portion of approximately 1 g of tendon was added for every 200 mL of diluted acetic acid (3% *v*/*v*) and incubated for 24 h at 8 ${}^{\circ }$C under moderate agitation with a magnetic stirrer. Next, the hydrolyzed collagen solution was filtered at room temperature with a gauze and centrifuged at 3000× *g* rpm for 30 min. Finally, the supernatants were collected by decantation and stored at −20 ${}^{\circ }$C.

### 2.3. Preparation of Collagen Membranes

The preparation of thin collagen membranes was carried out by neutralizing the stock collagen solution, previously prepared, with sodium hydroxide. A preliminary test was performed to determine the volume of soluble collagen and NaOH (1N) necessary to reach the isoelectric point: pH value in the range of 6.0–7.0. Therefore, the determined amounts of collagen and NaOH solutions were mixed, stirred quickly, and added to the Petri dishes immediately. The gelation of the solution was performed at room temperature, avoiding bubbles’ formation.

Subsequently, the resulting gels were washed three times with a phosphate buffer solution (20 mM) at pH 7.4. Finally, the gel samples were lyophilized to get the membranes.

### 2.4. Simulated Body Fluid Preparation

The simulated body fluid (SBF) was prepared by following the recipe proposed by T. Kokubo and H. Takadama [15]. First, to prepare 1000 mL of SBF, 700 mL of distilled water was poured into a 1000 mL plastic beaker with a stirring bar. The beaker was placed in a water bath on a magnetic stirrer, and the solution was heated until it reached 36.5 ${}^{\circ }$C. The first eight reagents shown in Table 1 were added to the solution in the order stated, attempting to completely dissolve one reagent at a time before proceeding to the next and maintaining the temperature at 36.5 ${}^{\circ }$C.

After dissolving the eight reagents, distilled water was added to reach a volume of 900 mL. A pH meter was then used to control the pH of the solution during the addition of the ninth (Tris) and tenth (HCl) reagent to avoid the precipitation of calcium phosphate. At a constant temperature of 36.5 ${}^{\circ }$C, the Tris reagent was slowly dissolved into the solution while monitoring the change in pH. After that, the pH of the solution was adjusted to 7.40 using small amounts of HCl (1M) and maintaining a constant temperature of 36.5 ${}^{\circ }$C. Then, the solution was poured into a 1000 mL volumetric flask previously washed with distilled water. The final volume of the solution was mixed and supplemented with distilled water.

### 2.5. Preparation of A and B Mineralization Solutions

Mineralization solutions A (rich in phosphate) and B (rich in calcium) were produced using a modified method for SBF preparation proposed by Bohner and Lemaitre [16].

First, deep cleaning of the materials was performed using a neutral detergent, dilute HCl solution, and distilled water. Then, using two volumetric flasks, 1000 mL of A and B solutions were prepared employing the reagents described in Table 2. Approximately 700 mL of distilled water was placed in a 2000 mL beaker and stirred with a magnetic stirrer at room temperature. Subsequently, the solid reagents were added, following the order indicated in the Table 2, waiting three minutes before adding the subsequent reagent to ensure its complete dissolution. The final volume was complemented with distilled water.

### 2.6. Mineralization Procedure

First, the dry collagen membranes were cut into 16 mm × 16 mm pieces to fit into the cells. Some of these pieces were pre-polarized before starting the mineralization process by placing them between two aluminum sheets and applying an electric field of 1 volt using an LCR meter U1731C from Agilent Technologies. Then, the mineralization procedure was conducted under two different modalities:

#### 2.6.1. Biomineralization in SBF-Reference Pattern Sample Preparation

A collagen sample was mounted in a plastic cell for in-vitro SBF incubation, as shown in Figure 1. The test piece was immersed in SBF for 45 min at a temperature between 36 ${}^{\circ }$C and 37 ${}^{\circ }$C by placing two 2 Ω resistors connected in series at a 12 V voltage on the cell’s outer surface.

#### 2.6.2. Electrophoretic Assisted Biomineralization

A double-diffusion system equipped with an electric field on the solution medium to promote the opposite migration of calcium and phosphate ions into collagen membranes acting as an organic template for mineral deposition was used. Biomineralization was performed in a dual container acrylic chamber, as shown in the Figure 2A. The dual container chamber was a system made up of two cells where the collagen pieces were placed in an intermediate space forming a barrier between them. The two reservoirs were filled in two different ways depending on the experiment: one with solution A and the other with solution B, or both with SBF. Moreover, to maintain the temperature of the solutions between 36 ${}^{\circ }$C and 37 ${}^{\circ }$C, the chamber incorporates eight 2 Ω resistors on the outer surface connected in series at a voltage of 12 V.

Additionally, each reservoir was equipped with a platinum wire acting as an electrode and connected to a DC power supply to provide a 1-volt electric field to the system. Then, the opposite migration of the positive and negative ions was expected to be promoted into and within the porous collagen membrane producing its biomineralization. The incubation times for the membranes were 10 and 45 min. After that, the mineralized samples were collected, washed with distilled water, and then dried at room conditions.

Finally, six experiments were performed to study the influence of the electric field, time, mineralization medium and membrane pre-polarization in the mineralization of the collagen. The convention used to label the samples consisted of a letter code as follows: (1)$$\chi -t_{\alpha }P_{\beta }E_{\gamma }$$
χ indicates the medium where the membrane was incubated. $t_{\alpha }$ indicates the incubation time that can be 10 or 45 min and is specified with the α index. $P_{\beta }$ refers to the pre-polarization of the collagen membrane before the mineralization process: when β = 0, the membrane was not pre-polarized, and when β = 1, the membrane was pre-polarized. $E_{\gamma }$ refers to the electric field: when γ = 0, the membrane was incubated in SBF without an electric field. Otherwise, when γ = 1, the membranes were incubated in the dual container chamber with the application of an electric field. Table 3 shows the pattern sample and the six experiments performed under different conditions specified by a letter code presented above.

### 2.7. Characterization of Structure and Properties

Mineralized collagen samples were washed with distilled water and dried under ambient conditions before performing the characterization procedures. The identification of the functional groups and the formation of HA inside collagen fibrils were evaluated with Fourier Transform Infrared Spectroscopy (FTIR) using a Perkin Elmer Spectrum 100 spectrometer coupled with a Microscope Spotlight 200. The spectra were recorded in the range of 400 to 4000 cm${}^{-1}$ with 4 cm${}^{-1}$ resolution. Additionally, it is known that analyzing the profile of the amide I band in the FTIR spectra can yield valuable information regarding changes in the secondary structure and composition of collagen caused by mineralization [17]. Hence, a curve-fitting treatment was carried out to deconvolute the amide I band using Gaussian function in OriginPro 2018 SR1 software. To accomplish this, a linear baseline was assumed, and the relative position and number of the band components were estimated based on the calculation of the second derivative function and previously reported studies [17,18].

To complement the FTIR studies, the Raman spectra of the samples were obtained using a LabRAM HR Evolution HORIBA spectrophotometer at an excitation wavelength of 633 nm with a 4 cm${}^{-1}$ resolution. Steady-state photoluminescence measurements were conducted at room temperature using a PL imaging setup consisting of a diode laser and an HR4000CG Ocean Optics spectrometer with an excitation wavelength of 405 nm. The conductance and capacitance of pure collagen and the mineralized samples were recorded using an LCR meter (Agilent Technologies U1731C) at room temperature for four different frequencies: 10 kHz, 1 kHz, 120 Hz, and 100 Hz. For that, the collagen membranes were connected to the LCR meter through sheets of aluminum foil attached at each end of the samples.

## 3. Results

### 3.1. FTIR and Raman Spectral Features

The FTIR analysis of the pure dry collagen and the six different mineralized samples are shown in Figure 3, and the band assignments and frequencies of the characteristic absorption peaks are given in Table 4. Moreover, the complementary Raman analysis of the samples are shown in Figure 4, while the general band assignments are given in Table 5.

#### 3.1.1. Pure Collagen

The pure collagen film exhibited a characteristic IR spectrum identifiable from the amide I, amide II, and amide III absorptions (Figure 3a). The amide I band at 1664 cm${}^{-1}$ resulted from the stretching vibration of the carbonyl bond (C=O) from the peptide group, while the amide II band at 1560 cm${}^{-1}$ is associated with the CH_2_ bending vibrations, C-N stretching vibrations and N-H bending vibrations. The Figure 5A illustrates the curve-fitting analysis of the amide I profile in the pure collagen FTIR spectrum. The analysis revealed four components derived from the heterogeneity of the peptide C=O group in the collagen triple helix [18]. The main component, representing the prevalence of the α-helical structure, is observed at approximately 1671 cm${}^{-1}$ (27.95% area). Additionally, the presence of random coils at around 1642 cm${}^{-1}$ (49.38%), turns at 1693 cm${}^{-1}$ (13.34%), and β-sheets at 1624 cm${}^{-1}$ (9.33%) can be observed. However, no components associated with the α helical structure are present, suggesting their presence in trace amounts or their absence in the sample.

Moreover, the amide III band at 1242 cm${}^{-1}$ arises from the N-H bending coupled with the CH_2_ wagging and C-N stretching vibrations. The wavenumbers at 1456 cm${}^{-1}$ and 1404 cm${}^{-1}$ are attributed to the stretching of the CH_2_ and CH_3_ bonds respectively according to previous studies [28]. The peak at 2965 cm${}^{-1}$ is ascribed to CH_3_ symmetric stretching while the broad band at approximately 3000–3400 cm${}^{-1}$ is assigned to OH vibration. Then, the described FTIR peaks agree with the reported literature which confirms that the extraction method did not damage the collagen structure.

On the other hand, Figure 4c shows the Raman spectrum obtained from pure dry collagen film. The amide I band centered at 1669 cm${}^{-1}$ arises from the vibration of the C=O functional group in the peptidic bond of the Gly-X-Y tripeptide sequence [29]. The amide III band appeared at 1256 cm${}^{-1}$ corresponding to the vibration coupling of the C–N stretching and N–H bending deformation. The medium/high bands at 1452 and 1005 cm${}^{-1}$ arise from the CH_2_ or CH_3_ deformations and the phenylalanine-breathing mode respectively [33]. Moreover, the bands assigned to the proline ring were located at 864 cm${}^{-1}$, arising from the νCC ring modes, and 938 cm${}^{-1}$, which belongs to the C-C stretching of the backbone constituted by the tripeptide sequence of collagen [29]. Finally, the peak at 818 cm${}^{-1}$ is attributed to the C-O-C stretching, while the S-S vibration that appeared at 530 cm${}^{-1}$ arises from the disulfide bridges of the 7S domains [29].

#### 3.1.2. Synthesized HA

Figure 6A presents the FTIR spectrum of pure hydroxyapatite previously synthesized by chemical precipitation, where the characteristic functional groups (PO_4_)${}^{3-}$, (CO_3_)${}^{2-}$, OH^−^ and (HPO_4_)${}^{2-}$ were identified. The apatite calcium phosphate bands were observed at 471 cm${}^{-1}$ (ν_2_), 564–603 cm${}^{-1}$ (ν_4_ O-P-O asymmetric bend), 962 cm${}^{-1}$ (ν_1_ P-O symmetric stretch), and 1033–1093 cm${}^{-1}$ (ν_3_ P-O asymmetric stretch). Hydroxyapatite usually presents carbonate groups due to two types of carbonate substitution (A and B). A-type consists of the direct substitution of OH^−^ with (CO_3_)${}^{2-}$ while B-type involves substitution of a (PO_4_)${}^{3-}$ tetrahedral group with (CO_3_)${}^{2-}$ because of charge compensation [23]. Then, the weak peaks at 1421 and 1455 cm${}^{-1}$ confirmed the presence of B and A carbonate ions in the synthetized hydroxyapatite sample.

The stretching mode at 3567 cm${}^{-1}$ and bending vibration at 630 cm${}^{-1}$ dominantly arise from the hydrogen-bonded ions OH^−^ (hydroxyl groups), while the weak band at 875 cm${}^{-1}$ belongs to both (HPO_4_)${}^{2-}$ and (CO_3_)${}^{2-}$ ions [35].

Moreover, the Raman analysis of hydroxyapatite is presented in Figure 6B. The spectrum showed characteristic bands of (PO_4_)${}^{3-}$ functional group: ν_1_ symmetric stretching mode of the P-O bond at 960 cm${}^{-1}$, ν_2_ bending mode of the P-O-P linkage at 434 cm${}^{-1}$, ν_3_ asymmetric stretching mode of the P-O bond at 1087 cm${}^{-1}$, 1072 cm${}^{-1}$, and 1044 cm${}^{-1}$, and ν_4_ bending mode of the P-O-P linkage at 589 cm${}^{-1}$ [36,37]. The other expected bands near 450 cm${}^{-1}$ (ν_2_), 580 cm${}^{-1}$ (ν_4_), and 610 cm${}^{-1}$ (ν_4_) arising from the bending modes of the P-O-P have weak intensities and were not detected. The reason why these bending and stretching bands present weak intensities in the Raman spectrum may be attributed to the water vibrational modes, as has been discussed in previous studies [38].

### 3.2. Mineralized Collagen

#### 3.2.1. Raman Spectroscopy

The Raman spectra of the mineralized samples, Figure 4, show some of the characteristics bands of collagen, amide I and III, that seem to be shifted after biomineralization. The amide I band (stretching C=O), centered at 1669 cm${}^{-1}$ in pure collagen, was found around 1667–1677 cm${}^{-1}$ in the mineralized collagen samples. The amide III band (C–N stretching and bending N–H) was found in collagen at 1256 cm${}^{-1}$ and between 1245–1254 cm${}^{-1}$ after mineralization. Additionally, the spectral features of hydroxyapatite were observed in mineralized collagen at around 937–947 cm${}^{-1}$ from ν_1_ (PO_4_)${}^{3-}$, 550–574 cm${}^{-1}$ from ν_4_ (PO_4_)${}^{3-}$, 434–445 cm${}^{-1}$ from ν_2_ (PO_4_)${}^{3-}$, and those at 1030–1045 and 1064–1087 cm${}^{-1}$ from ν_3_ (PO_4_)${}^{3-}$ vibrations. Here, the ν_1_ (PO_4_)${}^{3-}$ vibration band seems overlapped with the proline ring’s C-C stretching band, located at 938 cm${}^{-1}$ in pure collagen, which could be the cause of its red-shifting from 960 cm^−1^ in HA spectrum to 941–946 cm${}^{-1}$ in all the mineralized samples. Then, deconvolution procedures were performed using the Gaussian function in OriginPro 2018 software to correctly determine the mineral to matrix ratio (phosphate ν_1_/amide I ratio) of the samples (Table 6).

Generally, all the peaks shifting in the mineralized collagen spectrum arise from the interactions between the hydroxyapatite crystals and the collagen structure. The most critical interactions in Collagen/HA materials are the hydrogen bonding and electrostatic interactions between hydroxyapatite grains, different collagen structures, and the organic and inorganic phases due to opposite charges [22].

The hydrogen bonds can occur inside collagen molecules. Interactions between the three constituents polypeptides (Gly-X-Y) [22], and between collagen and HA molecules take place, specifically between the oxygen atom of the HA and the hydrogen atom of collagen [28]. Otherwise, the electrostatic interaction in collagen/HA materials occurs between the negative carboxylic groups from collagen with the calcium cations from hydroxyapatite [28]. This is evidenced by the clear difference between the C-O and C=O band (amide I) in pure and mineralized collagen spectra.

#### 3.2.2. FTIR Analyses of Mineralized Samples under Different Experimental Conditions

The FTIR spectra of different mineralized samples in Figure 3, exhibited characteristic bands of HA and collagen with modifications in their intensities and wavenumbers resulting from the mineralization process. First, the typical bands of collagen such as amide I, II and III were observed at around 1637–1670, 1546–1556, and 1242 cm${}^{-1}$, respectively. The ν_3_ vibration mode from phosphate groups expected at 1093 cm${}^{-1}$ in pure HA had been red-shifted to ∼1081 cm${}^{-1}$ for the mineralized samples. Simultaneously, the band at 564 cm${}^{-1}$ (associated with ν_4_ O-P-O bend) characteristic of the HA phase spectrum was found around 550–576 cm${}^{-1}$ in the mineralized samples. The intensity of the OH stretching band (3200–3500 cm${}^{-1}$) in the mineralized membranes appeared much lower than that of pure collagen. Besides, the OH bending vibration presented a large blue shifting for mineralized collagen (652–672 cm${}^{-1}$) compared to hydroxyapatite (630 cm${}^{-1}$). This shifting may be the result of calcium ions chelating with hydroxyl groups growing into nanoscale particles on the surface of collagen fibrils [25].

In order to in-depth analyze the effect of time, electric field and polarization on the mineralization of collagen a detailed discussion is presented based on the changes observed in Amide bands by FTIR and Raman spectroscopy. The profile shape of the amide 1 contains different peaks, due to the heterogeneity structure of the peptides groups of collagen. Therefore deconvolution of the amide 1 band should be a valuable tool for examining the structural changes occurring during mineralization under the different experiments carried out. The deconvolution was performed in the range of 1600–1720 cm${}^{-1}$ (Figure 5) The centred band vibrations of α helix (1650–1660 cm${}^{-1}$), β sheets (1610–1642 cm${}^{-1}$), random coils (1642–1650 cm${}^{-1}$) and beta turns (1660–1700 cm${}^{-1}$) were assigned according to the literature [39,40,41,42,43] and are shown in the deconvoluted profile (Figure 5).

## 4. Discussion

### 4.1. Time Effect

During the mineralization process several changes can be observed in the FTIR peak of amide 1. These changes are indicative of the structural modifications occurring in collagen as it interacts with the mineral phase. The effect of the time in the collagen mineralization was evaluated by examining the differences between two collagen/HA samples prepared under the same conditions but at different incubation times: AB-$t_{10}P_{0}E_{1}$ and AB-$t_{45}P_{0}E_{1}$.

The FTIR spectra, Figure 3a,b, present some changes in the profile of the amide I and II peaks as a function of time: the amide I band in AB-$t_{10}P_{0}E_{1}$ showed three overlapped peaks while AB-$t_{45}P_{0}E_{1}$ only showed two. This phenomenon may result due to the difference in hydration level of the membranes. The deconvoluted spectrum of the amide I band showed three main peaks corresponding to α-helices, β-sheets and a small peak assigned to beta turns visible under complete dehydration of the collagen (Figure 5A) [43].

Moreover, the FTIR mineral to matrix ratio of AB-$t_{45}P_{0}E_{1}$ (0.39) and AB-$t_{10}P_{0}E_{1}$ (0.29) were almost similar with a slight increment in AB-$t_{45}P_{0}E_{1}$ that may suggest a higher mineralization degree with time. A shift in the location of the alfa helix towards lower wavenumbers and an increase in the beta sheets area (Figure 4c,d may be due to the increase in mineralization degree with time). Analogous studies show that the mineralization degree increases with the conditioning time [44]. These shifts may indicate changes in the hydrogen bonding and conformation of the collagen peptide backbone due to the interaction with the mineral phase.

Otherwise, Raman spectra of two samples AB-$t_{10}P_{0}E_{1}$ and AB-$t_{45}P_{0}E_{1}$ were presented in Figure 4d,e respectively. The samples spectra showed no peak shifting differences but some intensity changes. For instance, it was observed a widening of the amide I and III bands at 45 min compared to 10 min suggesting an increase in the interactions between the mineral and organic phase and higher mineral deposition. Then, according to Table 6, the mineral to matrix ratio was higher in AB-$t_{45}P_{0}E_{1}$ (0.53) than in AB-$t_{10}P_{0}E_{1}$ (0.46). The difference between ratios was larger than the observed in the FTIR analysis, and confirms that the mineralization degree increases with time.

### 4.2. Electric Field Effect

To evaluate the effect of the electric field two non-pre-polarized collagen membranes, incubated for 45 min in SBF with (SBF-$t_{45}P_{0}E_{1}$) and without (SBF-$t_{45}P_{0}E_{0}$) the application of 1 volt electric field on the solution media, were analyzed.

Moreover, the FTIR mineral to matrix ratioIn the FTIR spectra, it was observed that the shape profile of the amide I band for the sample without the electric field SBF-$t_{45}P_{0}E_{0}$ (standard sample) was different from that observed in the spectrum of the sample treated with the electric field. The deconvoluted spectra (Figure 5E,F) showed an increase in α-helices area and a shift towards lower wavenumber with mineralization time under the application of the electric field, as well as a decrease in random coils, suggesting a higher degree of order with mineralization under the application of the electric field.

This is expected since the application of the electric field in the solution increase the mobility of calcium and phosphate ions and therefore an increase in the mineratization degree is obtained. On the other side, it has been proven that the electric field application can induce conformational changes in proteins. Such changes may lead to intramolecular charge separations or reorientation of a charged group inside a matrix affecting the functional stability and properties of the protein [45]. Collagen is a protein whose mineralization is believed to occur in the gap regions of the fibrils. These sites are preferred for mineral nucleation because their positive net charges allow extending crystals across microfibrillar collagen spaces [46]. Therefore, applying the electric field to the incubation medium could change the structural arrangement of the charges in the collagen matrix. As a result, this could affect the interactions between the mineral and the collagen matrix, specifically those between the carboxyl group of collagen and the calcium ions of HA. However, further studies and analysis are needed to verify if the electric field can modify the structure and conformational organization of collagen fibrils and induce changes in the orientation and shape of the HA nanocrystals, which could have a significant impact on mineralization.

Furthermore, the FTIR semi-quantitative analysis (Table 6) confirmed that the mineral to matrix ratio of the samples had increased under the presence of the electric field. The average phosphate peak over the amide I peak ratio of SBF-$t_{45}P_{0}E_{1}$ (0.14) was higher than that of SBF-$t_{45}P_{0}E_{0}$ (0.11). These results suggest that applying an electric field on the solution media increases the ions migration rate into and within the collagen membrane boosting the mineral formation, as proven in analogous studies [6,7].

Otherwise, the Raman analysis showed that the amide I and III bands centered at 1677 cm${}^{-1}$ and 1247 cm${}^{-1}$ in SBF-$t_{45}P_{0}E_{0}$ were shifted to 1671 cm${}^{-1}$ and 1254 cm${}^{-1}$ in SBF-$t_{45}P_{0}E_{1}$. The ν_1_ (PO_4_)${}^{3-}$ vibrational mode in SBF-$t_{45}P_{0}E_{1}$ exhibited a shoulder peak attributed to the proline vibration as it was previously suggested, which was no observed in the SBF-$t_{45}P_{0}E_{0}$ spectrum. Then, with the application of the electric field the characteristics collagen bands in the SBF-$t_{45}P_{0}E_{1}$ spectrum are better resolved. As well as, the Raman semi-quantitative analysis shows that the mineral to matrix ratio of SBF-$t_{45}P_{0}E_{1}$ (0.45) was much higher than that of SBF-$t_{45}P_{0}E_{0}$ (0.19). This confirmed that the mineralization degree increases under the application of an electric field, in agreement with the FTIR results.

### 4.3. Polarization Effect

The polarization effect was evaluated in collagen/HA samples spectra obtained from pre-polarized and non-pre-polarized collagen membranes incubated for 10 min (AB-$t_{10}P_{1}E_{1}$ and AB-$t_{10}P_{0}E_{1}$), and 45 min (AB-$t_{45}P_{1}E_{1}$ and AB-$t_{45}P_{0}E_{1}$) in solutions AB under electric field of one volt applied to the media.

For the collagen membranes incubated for 10 min (Figure 3g,h), the mineral to matrix ratio of the polarized sample (AB-$t_{10}P_{1}E_{1}$ = 0.18) was substantially lower than that of the non-polarized matrix (AB-$t_{10}P_{0}E_{1}$ = 0.29). However, at 45 min (Figure 3c,e), the mineral to matrix ratios were similar for both samples (AB-$t_{45}P_{1}E_{1}$ = 0.29 and AB-$t_{45}P_{0}E_{1}$ = 0.30). The FTIR deconvoluted spectra (Figure 5G,H) showed an increase in α-helices formation in the pre-polarized sample and some decrease in turns for the same mineralization degree. The effects of the polarization on collagen can be attributed to the alignment and reorientation of collagen molecules and changes in the dipole orientation and hydrogen bonding interactions within the collagen matrix, leading to alterations in the secondary structure [47].

On the other hand, the Raman studies showed that at 45 min, the ν_1_ (PO_4_)${}^{3-}$ vibrational mode was centered at 946 cm${}^{-1}$ in the pre-polarized sample and at 937 cm${}^{-1}$ in the non-pre-polarized collagen membrane. Also, the ν_3_ and ν_4_ (PO_4_)${}^{3-}$ vibrational modes (1045 and 574 cm${}^{-1}$) are better resolved in the pre-polarized than in the non-pre-polarized sample (566 and 1030 cm${}^{-1}$), indicating a higher mineral concentration. The mineral to matrix ratio of AB-$t_{45}P_{1}E_{1}$ (0.85) was much higher than that of AB-$t_{45}P_{0}E_{1}$ (0.53). Therefore, an increase of the collagen mineralization degree was observed in the Raman spectra when the membranes were pre-polarized for 45 min, due to the higher sensitivity of the latter technique. Our hypothesis is that the polarization of collagen membranes causes a change in the orientation of electrical charges along its structure, creating an electrical gradient that attracts phosphate and calcium ions to the membrane surface during the biomineralization. As a result, there is an increase in the rate of formation of hydroxyapatite crystals and on the degree of mineralization.

When an electric field is applied to a collagen membrane, it can induce the alignment of collagen molecules and fibers in a specific direction, creating a polarized environment. This can result in the alignment of charged amino acid residues within the collagen molecules, which can affect their behavior and interactions with other molecules, including mineral ions. Specifically, the application of an electric field can induce the alignment of charged groups in the collagen molecules, such as carboxylate and amine groups, which can interact with mineral ions such as calcium and phosphate. These interactions can facilitate the deposition and organization of mineral ions, leading to the formation of a mineralized matrix that is integrated with the collagen fibers. Overall, the alignment of charged groups in collagen molecules can play a critical role in the polarization of collagen and the subsequent mineralization process. The specific types of charged groups that are aligned will depend on the amino acid composition of the collagen molecule, as well as the strength and direction of the applied electric field. In addition to the alignment of charged groups, the application of an electric field can also induce conformational changes in collagen molecules. Collagen molecules have a characteristic triple-helical structure, but the application of an electric field can alter the orientation of the helices, leading to changes in the overall structure and alignment of the collagen fibers. These changes in the structure can also affect the behavior of mineral ions and the formation of mineralized matrices. Furthermore, the polarization of collagen can also affect the behavior of cells, such as osteoblasts and chondrocytes, which are responsible for the deposition of mineral ions and the formation of new tissue. The aligned collagen fibers can provide a directional signal that guides cell migration and tissue formation, leading to more organized and functional tissue structures.

### 4.4. Photoluminescence Studies

Figure 7 presents the photoluminescence spectra of pure collagen and the seven prepared collagen/HA hybrid materials. Under the excitation of a 405 nm light source, pure collagen and its mineralized samples exhibited a strong emission at around 498 nm with a fast luminescence quenching. This fast quenching phenomenon is a result of the short lifetime (<10^−5^ s) of the fluorescent pathway in collagen light emission [48]. Additionally, a small broadband at about 670 nm was also observed in the PL spectra of the mineralized samples. This band may indicate the presence of some HA precursors crystals in the collagen matrix, such as amorphous calcium phosphate (ACP) [49].

Overall, based on the quantitative Raman and FTIR analysis of the collagen/HA materials, most of the PL results suggest that the luminescence intensity was stronger at higher mineralization degrees. The luminescence was improved upon the application of the electric field since SBF-$t_{45}P_{0}E_{1}$ (3 pink curve) exhibited a higher intensity than SBF-$t_{45}P_{0}E_{0}$ (1 cyan curve). Moreover, the membrane pre-polarization also improved the luminescence properties of the mineralized samples at 10 min of incubation. The pre-polarized sample (8 red curve AB-$t_{10}P_{1}E_{1}$) exhibited a higher intensity than the non-pre-polarized one (7 green curve AB-$t_{10}P_{0}E_{1}$). Therefore, just like the Raman and FTIR studies, the PL results indicate that the electric field application and the membrane pre-polarization promote better mineralization of the collagen, improving its luminescence properties.

Generally, the intrinsic fluorescence of proteins arises from the aromatic amino acid groups present in their structure, including phenylalanine, tryptophan, and tyrosine residues [50]. In the case of collagen, its luminescence comes exclusively from tyrosine due to the lack of phenylalanine and tryptophan residues, and the mineralization process tends to improve it. Previous studies have shown that the addition of hydroxyapatite to collagen results in a material with an enhanced fluorescence and brightness, which remained constant due to the presence of phosphate components with phosphorescent nature [48]. However, even though the luminescence intensity was enhanced with the mineralization degree, the hybrid materials exhibited a fast quenching and a fluorescence peak with a shape and absorption maximum similar to the collagen. This phenomenon could be caused by the reduced concentration of HA and the prevalence of collagen in the mineralized samples.

Otherwise, the luminescence properties of some mineralized samples were observed to decrease with time and pre-polarization at 45 min of incubation. For collagen membranes incubated in solutions A-B under the application of the electric field, a substantial decrease of the luminescence intensity was showed at 45 min (4 black curve AB-$t_{45}P_{0}E_{1}$) compared to 10 min (7 green curve AB-$t_{10}P_{0}E_{1}$). Similarly, for membranes incubated for 45 min, the pre-polarized sample (2 purple curve AB-$t_{45}P_{1}E_{1}$) exhibited a lower luminescence intensity than the non-pre-polarized one (4 black curve AB-$t_{45}P_{0}E_{1}$). This phenomenon could be due to the growth of the mineral crystals inside and on the surface of the collagen membranes could cover the tyrosine residues [51], reducing the luminescence properties of some collagen/HA samples.

### 4.5. Electrical Characterization Study

The electric properties plotted as a function of the frequency of pure collagen and the mineralized samples are shown in Figure 8. The membranes’ capacitance (*C*) and conductance (G) were measured at room temperature. The dielectric constant (ε_*r*_) of the samples was calculated from the following formula: (2)$$\begin{array}{cc} \varepsilon_{r} & =\frac{Cd}{\varepsilon_{0}A} \end{array}$$
where *d* is the thickness of the membrane, ε_0_ is the permittivity of vacuum, and *A* is the sample area in contact with the aluminum layers. Otherwise, the conductivity (σ) was calculated from the measured conductance using the following equation: (3)$$\begin{array}{cc} \sigma & =\frac{Gd}{A} \end{array}$$

Figure 8B,D indicate that the dielectric constant and capacitance decrease with frequency increase. This results from the orientation of the dipoles in the material along the direction of the applied field at low-frequency values. However, with the increase of the frequency, the dipoles do not have enough time to orient with the field, decreasing the polarization and the dielectric constant of the compound [52].

Figure 8D shows that the dielectric constant tend to be smaller for mineralized samples than pure dry collagen. In addition, a reduction of the dielectric properties was also observed with the increase of the degree of mineralization. For samples incubated under the effect of the electrical field and those pre-polarized, which have been proven to have a higher mineralization degree, their dielectric constants were lower than their counterparts. However, there was an exception for samples incubated for 45 min (AB-$t_{45}P_{0}E_{1}$) and 10 min (AB-$t_{10}P_{0}E_{1}$) where the dielectric properties were higher for the sample with higher mineral content (AB-$t_{45}P_{0}E_{1}$).

The dielectric properties in a solid material depend upon the movement of the charges inside the molecules. The dielectric constant increases with the material conductivity, and the hydrogen bonds present in hydroxyapatite (HA) and collagen enhance the movement of charges in the presence of an electric field [53]. Then, the decrease in the dielectric constant of the mineralized collagen compared to pure collagen may be due to the breaking and decrease of the hydrogen bonds inside the phases of the mineralized material (organic and mineral). In addition, a higher mineral deposition on and within the collagen matrix might cause a decrease in the segmental rotation [53]. The reduction of freedom of the side chains in mineralized collagen compared to the pure sample would decrease the total dielectric permittivity of the material as it has been proven in similar studies performed in bones [53,54].

## 5. Conclusions

The electric field-assisted double diffusion system turned out to be a successful and novel method for collagen biomineralization. The electric field application significantly accelerates the formation of HA while the membrane pre-polarization proved to induce faster mineralization. The FTIR and Raman structure analysis confirmed the preservation of the collagen structure and the presence of conformational changes with biomineralization under the different experimental conditions. The electrical properties of the mineralized collagen, including the dielectric constant and conductivity, were reduced with the increase of the mineral deposition. Otherwise, the collagen’s photoluminescence improved upon biomineralization with an intensity increment at higher mineralization degrees which could be a useful method for monitoring bone implants through bioimaging.

The mechanism of mineralization in electrically polarized collagen proposed involves the alignment of collagen fibers in a specific direction, creating a polarized environment that is favorable for the deposition and organization of mineral ions. This process is critical for the formation of mineralized matrices that are integrated with the collagen fibers, and it can be used in tissue engineering applications to promote the growth and regeneration of new tissue. Therefore, this research presents a new perspective on the use of polarization to enhance the mineralization of collagen membranes. The findings of this study may contribute to a better understanding of the mechanisms involved in collagen biomineralization and the potential of using electric fields to drive mineral deposition, which could be valuable for future applications in manufacturing biomimetic bone-like materials. 

## Figures and Tables

**Figure 1 polymers-15-03121-f001:**
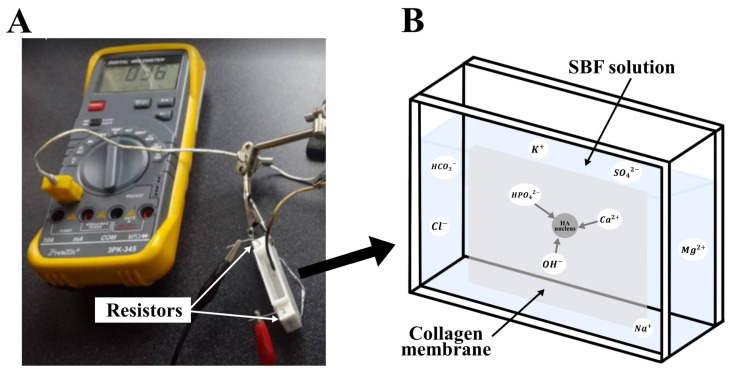
Experimental setup for in-vitro collagen biomineralization in SBF (**A**) photograph of the actual cell, and (**B**) its schematic representation.

**Figure 2 polymers-15-03121-f002:**
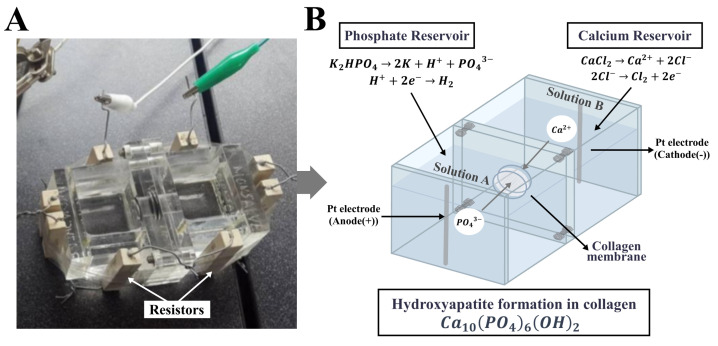
(**A**) Photograph of the dual container acrylic chamber for collagen biomineralization, and (**B**) schematic illustration of the double-diffusion system in it.

**Figure 3 polymers-15-03121-f003:**
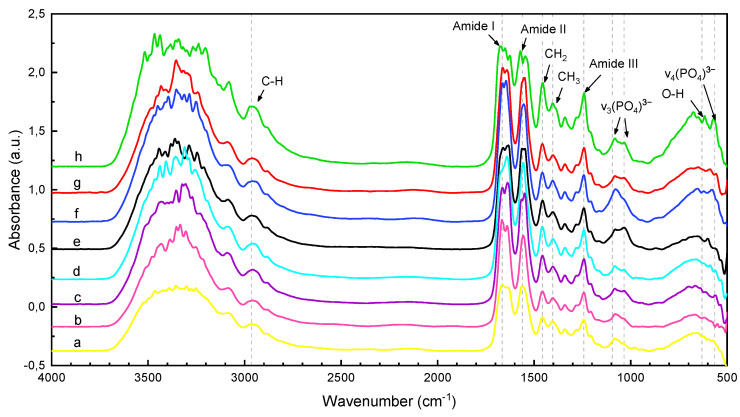
FTIR spectra of (**a**) pure dry collagen and the synthesized collagen–hydroxyapatite samples: (**b**) SBF-$t_{45}P_{0}E_{1}$, (**c**) AB-$t_{45}P_{1}E_{1}$, (**d**) SBF-$t_{45}P_{0}E_{0}$, (**e**) AB-$t_{45}P_{0}E_{1}$, (**f**) SBF-$t_{10}P_{0}E_{1}$, (**g**) AB-$t_{10}P_{1}E_{1}$ and (**h**) SBF-$t_{10}P_{0}E_{1}$.

**Figure 4 polymers-15-03121-f004:**
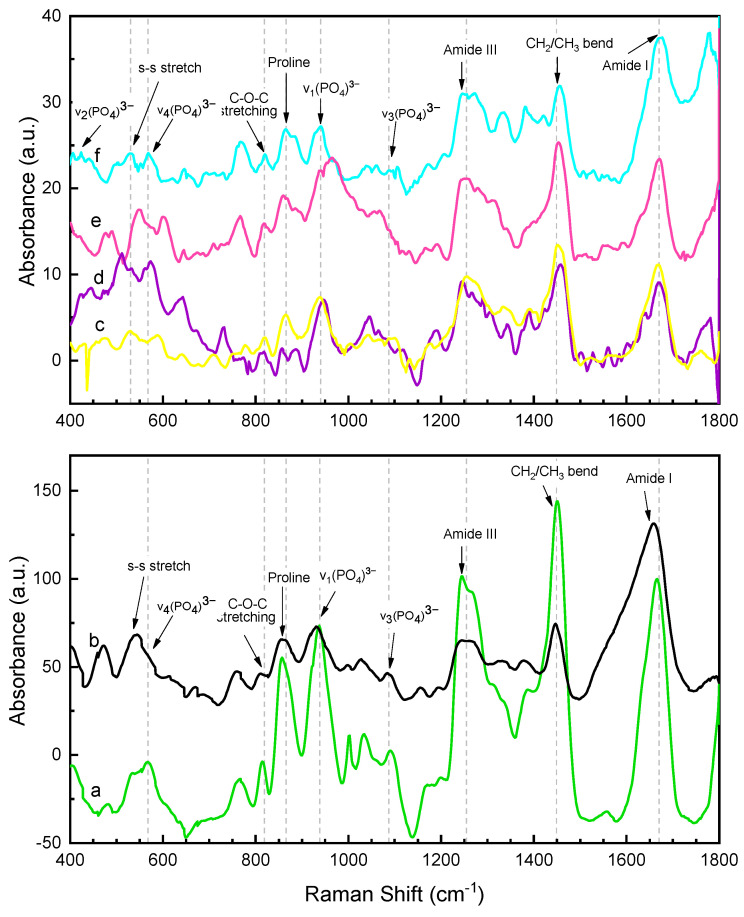
Raman spectra of pure dry collagen and the mineralized collagen samples, under the different experimental conditions: (**a**) SBF-$t_{10}P_{0}E_{1}$, (**b**) AB-$t_{45}P_{0}E_{1}$, (**c**) pure collagen, (**d**) AB-$t_{45}P_{1}E_{1}$, (**e**) SBF-$t_{45}P_{0}E_{1}$, and (**f**) SBF-$t_{45}P_{0}E_{0}$.

**Figure 5 polymers-15-03121-f005:**
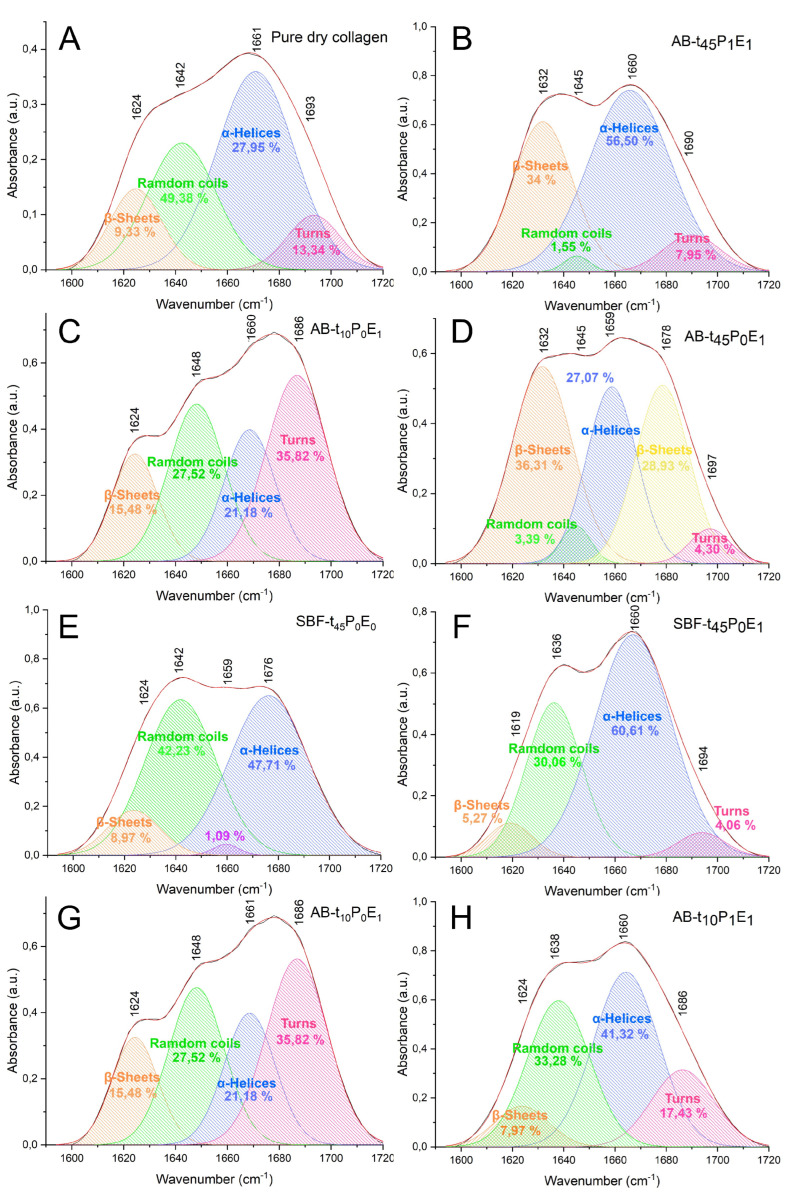
Curve-fitting analysis of the amide I profile on FTIR spectra of (**A**) pure dry collagen and the synthesized collagen–hydroxyapatite samples: (**B**) SBF-*t*${}_{45}P_{1}E_{1}$, (**C**) AB-*t*${}_{10}P_{0}E_{1}$, (**D**) AB-*t*${}_{45}P_{0}E_{1}$, (**E**) SBF-*t*${}_{45}P_{0}E_{0}$, (**F**) SBF-*t*${}_{45}P_{0}E_{1}$, (**G**) AB-*t*${}_{10}P_{1}E_{1}$, (**H**) AB-*t*${}_{10}P_{0}E_{1}$.

**Figure 6 polymers-15-03121-f006:**
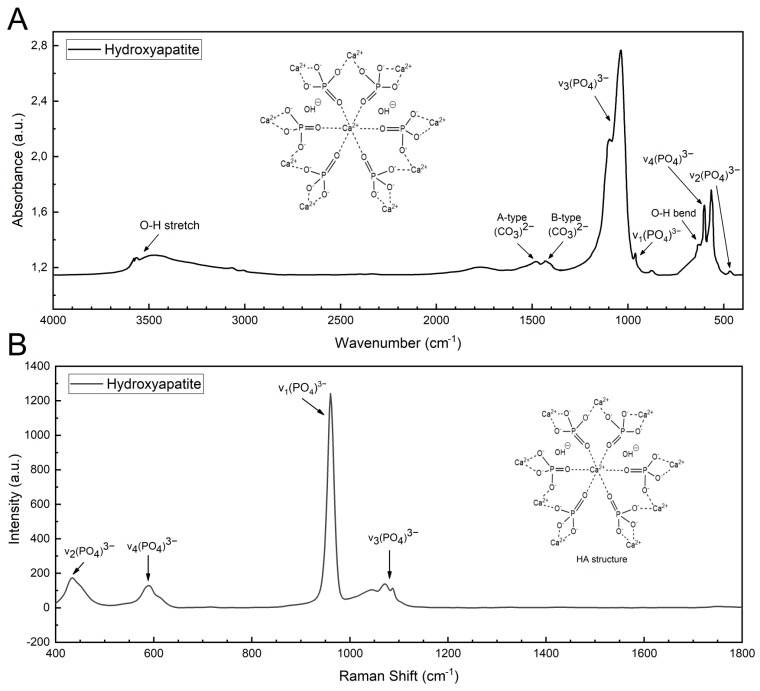
(**A**) FTIR and (**B**) Raman spectra of hydroxyapatite (HA).

**Figure 7 polymers-15-03121-f007:**
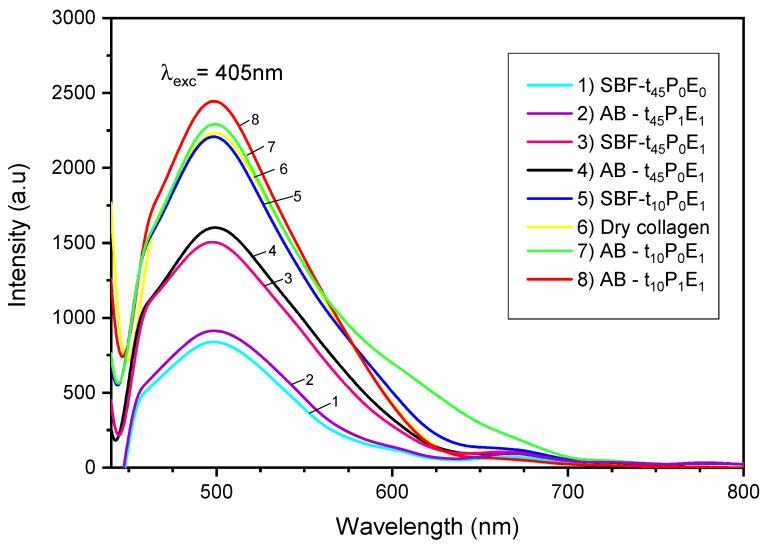
PL spectra of pure collagen and the collagen/HA samples.

**Figure 8 polymers-15-03121-f008:**
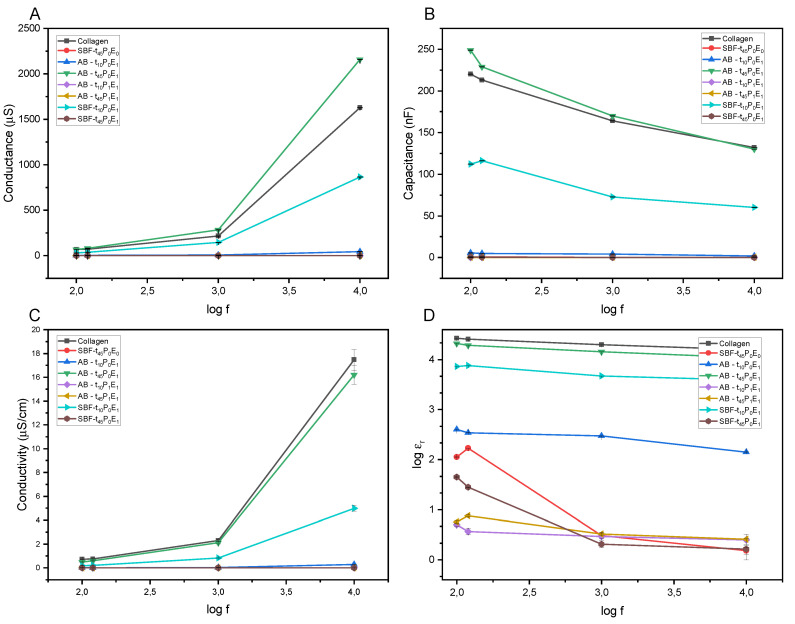
(**A**) Conductance, (**B**) capacitance, (**C**) conductivity and (**D**) dielectric constant of pure collagen and the collagen/HA samples.

**Table 1 polymers-15-03121-t001:** Preparation of The simulated body fluid (SBF).

Order	Reagent	Amount	Purity (%)
1	NaCl	8.035 g	99.5
2	NaHCO_3_	0.355 g	99.5
3	KCl	0.225 g	99.5
4	K_2_HPO4·3H_2_O	0.231 g	99.0
5	MgCl2·6H_2_O	0.311 g	98.0
6	HCl (1 M)	39 mL	–
7	CaCl_2_	0.292 g	95.0
8	Na_2_SO_4_	0.072 g	99.0
9	C_4_H_11_O_3_N (Tris)	6.118 g	99.0
10	HCl (1 M)	0–5 mL	–

**Table 2 polymers-15-03121-t002:** Preparation of A and B solutions.

Starting Materials	Sol.A (g/L)	Sol.B (g/L)	Purity (%)
NaCl	3.1065	3.1065	99.5
NaHCO_3_	2.974	–	99.5
KCl	0.225	–	99.5
K2HPO_4_·3 H_2_O	0.231	–	99.0
MgCl_2_·6 H_2_O	0.311	–	98.0
CaCl_2_·2 H_2_O	–	0.292	95.0
Na_2_SO_4_	0.072	–	99.0
HCl (mL/L)	0.425	0.425	–

**Table 3 polymers-15-03121-t003:** Collagen biomineralization experiments.

Experiment	Name
Pattern	SBF-*t*${}_{45}P_{0}$E${}_{0}$
1	AB-*t*${}_{10}P_{0}$E${}_{1}$
2	AB-*t*${}_{45}P_{0}$E${}_{1}$
3	AB-*t*${}_{10}P_{1}$E${}_{1}$
4	AB-*t*${}_{45}P_{1}$E${}_{1}$
5	SBF-*t*${}_{10}P_{0}$E${}_{1}$
6	SBF-*t*${}_{45}P_{0}$E${}_{1}$

**Table 4 polymers-15-03121-t004:** Peak positions and vibration modes of the bands in the FTIR spectra of the samples: $\mathrm{Exp}.\;1\;\left(\mathrm{AB}\text{-}t_{10}P_{0}E_{1}\right),\;$
$\mathrm{Exp}.\;2\;\left(\mathrm{AB}\text{-}t_{45}P_{0}E_{1}\right),\;$
$\mathrm{Exp}.\;3\;\left(\mathrm{AB}\text{-}t_{10}P_{1}E_{1}\right),$  
$\mathrm{Exp}.\;4\;\left(\mathrm{AB}\text{-}t_{45}P_{1}E_{1}\right),\;$
$\mathrm{Exp}.\;5\;\left(\mathrm{SBF}\text{-}t_{10}P_{0}E_{1}\right),\;\mathrm{and}\;$
$\mathrm{Exp}.\;6\;\left(\mathrm{SBF}\text{-}t_{45}P_{0}E_{1}\right)$
.

Vibration	Wavenumber (cm${}^{-1}$)									
	Collagen	HA	Ref. Pattern	Exp. 1	Exp. 2	Exp. 3	Exp. 4	Exp. 5	Exp. 6	Reference
ν _2_ ${\left(PO_{4}\right)}^{3-}$	–	471	–	–	–	–	–	–	–	[19,20]
ν _4_ ${\left(PO_{4}\right)}^{3-}$	–	564–630	562	568		556	563	576	550	[7,21,22]
O-H	–	630	633	672	654	688	666	652	667	[7,19,20]
${\left(HPO_{4}\right)}^{2-}$	–	875	–	–	–	–	–	–	–	[19,23]
ν _1_ ${\left(PO_{4}\right)}^{3-}$	–	962	–	–	–	–	–	–	–	[7,22,24]
ν _3_ ${\left(PO_{4}\right)}^{3-}$	–	1033	–	1033	1034	–	–	–	–	[7,21,22,24]
ν _3_ ${\left(PO_{4}\right)}^{3-}$	–	1093	1081	1081	1081	1080	1081	1077	1079	[7,24]
C-O ester	1081	–	–	–	–	–	–	–	–	[24]
Amide III	1242	–	1242	1240	1242	1241	1241	1242	1242	[21,22,24,25,26]
Amide C-N	1340	–	1340	1340	1340	1340	1340	1340	1340	[27]
CH_3_	1404	–	1402	1403	1403	–	–	–	1403	[28]
A-type ($CO_{3}{)}^{2-}$	–	1421	–	–	–	–	–	–	–	[23]
B-type ($CO_{3}{)}^{2-}$	–	1455	–	–	–	–	–	–	–	[19,23]
CH_2_	1456	–	1457	1455	1456	1457	1456	1457	1456	[24,28]
Amide II	1560	–	1554	1546	1546	1548	1549	1553	1556	[21,22,25,26]
Amide I	1664	–	1670	1670	1662	1663	1637	1645	1666	[21,22,24,25]
ν $C-H_{x}$	2965	–	2961	2969	2964	2959	2959	2958	2958	[27]

**Table 5 polymers-15-03121-t005:** Peak positions and vibration modes of the bands in the Raman spectra of the samples: $\mathrm{Exp}.\;1\;\left(\mathrm{AB}\text{-}t_{10}P_{0}E_{1}\right)$, $\mathrm{Exp}.\;2\;\left(\mathrm{AB}\text{-}t_{45}P_{0}E_{1}\right)$, $\mathrm{Exp}.\;3\;\left(\mathrm{AB}\text{-}t_{10}P_{1}E_{1}\right)$, $\mathrm{Exp}.\;4\;\left(\mathrm{AB}\text{-}t_{45}P_{1}E_{1}\right)$, $\mathrm{Exp}.\;5\;\left(\mathrm{SBF}\text{-}t_{10}P_{0}E_{1}\right)$, and $\mathrm{Exp}.\;6\;\left(\mathrm{SBF}\text{-}t_{45}P_{0}E_{1}\right)$.

Vibration	Wavenumber (cm${}^{-1}$)							
	Collagen	HA	Pattern	Exp. 1	Exp. 2	Exp. 4	Exp. 6	Reference
Amide I, C-C-N stretch	1669	–	1677	1666	1666	1671	1671	[24,29,30,31]
C-H bending	1452	–	1455	1451	1453	1458	1454	[30,31,32,33]
Amide III, C-N-H stretch	1256	–	1247	1246	1248	1245	1254	[24,30,31,32]
Phenylalanine	1005	–	–	–	–	–		[30,33]
$\left(PO_{4}\right){}^{3-}$ (v_1_)	–	960	941	937	937	946	941	[24,30,31,32]
$\left(PO_{4}\right){}^{3-}$(v_4_)	–	589	569	567	566	574	550	[24,31,32]
$\left(PO_{4}\right){}^{3-}$ (v_2_)	–	434	441	–	434	445	–	[24,30,32]
$\left(PO_{4}\right){}^{3-}$ (v_3_)	–	1087–1072–1044	1038–1087	1035–1089	1030	1045–1064	1067	[30,31]
νCC Proline	864		864	857	857	857	860	[29,31,32,34]
C-C Proline	938	–	–	–	–	–	–	[29,34]
C-O-C	818	–	819	814	816	–	818	[29]
S-S	530	–	531	–	–	–	–	[29]

**Table 6 polymers-15-03121-t006:** Mineral to matrix ratio of the different synthesized samples obtained from IR and Raman spectra.

Sample	IR Ratio	Raman Ratio
SBF-*t*${}_{45}P_{0}E_{0}$	0.11	0.19
AB-*t*${}_{10}P_{0}E_{1}$	0.29	-
AB-*t*${}_{45}P_{0}E_{1}$	0.30	0.53
AB-*t*${}_{10}P_{1}E_{1}$	0.18	-
AB-*t*${}_{45}P_{1}E_{1}$	0.29	0.85
SBF-*t*${}_{10}P_{0}E_{1}$	0.29	-
SBF-*t*${}_{45}P_{0}E_{1}$	0.14	0.45

## Data Availability

Not applicable.

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
