# Peer review of "Effect of the Electric Field on the Biomineralization of Collagen"

_polymers, 2023, doi:10.3390/polym15143121_

Round 1

Reviewer 1 Report

In this manuscript, Oriz et al. applied an external electric field to transport phosphate and calcium ions into a collagen gel matrix and subsequently form hydroxyapatite inside the gel matrix. This method could effectively enhance the mineralization degree. Several influences of pre-polarization, electric field, and processing time on the final mineralization degree were investigated. The properties of collagen composite were well characterized. It is worth publishing after minor revision; comments are the following:

(1) Authors should explain why they chose 10min and 45min incubation time. It is well known that the gel needs time to reach swelling equilibrium in aqueous. Does the incubation enough for the dry gel to reach equilibrium?

(2) It is better to explain why they set the applied voltage as 1V. Did they check the influence of applied voltage on the final mineralization degree?

(3) Authors should discuss the mechanism of pre-polarization affecting the mineralization.

Author Response

  • Authors should explain why they chose 10 min and 45min incubation time. It is well known that the gel needs time to reach swelling equilibrium in aqueous. Does the incubation enough for the dry gel to reach equilibrium?

Response 1. We have done experiments with conventional methods of immersion in SBF for periods between 1 to 6 weeks. In those experiments mineralization was very slow and only after three weeks we could see some initial mineralization. With these new experiments we wanted to demonstrate that applying an electric field for a very short time mineralization could be induced. Additionally, we wanted to explore the initial stages of this process and even equilibrium was not reached biomineralization was obtained after these short times. On the other side during therapeutic treatments for bone regeneration applying electric fields, the application periods are the range of times used here.

  • It is better to explain why they set the applied voltage as 1V. Did they check the influence of applied voltage on the final mineralization degree?

Response 2: in other set of experiments (ref J. Tipaz, 2018, undergrad thesis, Escuela Politécnica Nacional, Science Faculty, Quito, Ecuador) we have tried different voltages from 1 to 14 V. The result of those experiments was that high voltages resulted in a thick white layer of hydroxyapatite deposition. Since, we are interested in studying the first stages of mineralization, we choose 1 V to understand this process.

  • Authors should discuss the mechanism of pre-polarization affecting the mineralization.

             Response 3: The polarization process induces charges on the collagen molecule, creating dipoles that favour the Ca +2 and PO4 -3 ions interact with the charged collagen aminoacids.

The mechanisms of formation were expanded as follows: “When an electric field is applied to a collagen scaffold, it can induce the alignment of collagen molecules and fibers in a specific direction, creating a polarized environment. This can result in the alignment of charged amino acid residues within the collagen molecules, which can affect their behavior and interactions with other molecules, including mineral ions.

Specifically, the application of an electric field can induce the alignment of charged groups in the collagen molecules, such as carboxylate and amine groups, which can interact with mineral ions such as calcium and phosphate. These interactions can facilitate the deposition and organization of mineral ions, leading to the formation of a mineralized matrix that is integrated with the collagen fibers.

Overall, the alignment of charged groups in collagen molecules can play a critical role in the polarization of collagen and the subsequent mineralization process. The specific types of charged groups that are aligned will depend on the amino acid composition of the collagen molecule, as well as the strength and direction of the applied electric field.

In addition to the alignment of charged groups, the application of an electric field can also induce conformational changes in collagen molecules. Collagen molecules have a characteristic triple-helical structure, but the application of an electric field can alter the orientation of the helices, leading to changes in the overall structure and alignment of the collagen fibers. These changes in structure can also affect the behavior of mineral ions and the formation of mineralized matrices.

Furthermore, the polarization of collagen can also affect the behavior of cells, such as osteoblasts and chondrocytes, that are responsible for the deposition of mineral ions and the formation of new tissue. The aligned collagen fibers can provide a directional signal that guides cell migration and tissue formation, leading to more organized and functional tissue structures’.

Reviewer 2 Report

1. Introduction: Incomplete, need to elaborate more about previous studies, major scientific gaps and novelty of this study.

2. Materials and Methods: The source, manufacturer's details are missing throughout. 

2.2. Preparation of collagen membranes  The preparation of thin collagen gels: Gel or membrane

Need to prove that the Electric field on the Biomineralization of Collagen did not change the molecular pattern of collagen, which directly affect the behaviour and biological performance. Through, the authors performed FTIR and Raman spectra, its better to confirm the changes in secondary structure, amino acids composition and biocompatibility of collagen using invitro studies. 

Results: Unit y-Axis label is missing in all Figures, revise

The heading "Discussion" is missing.

4. Conclusions: Incomplete, describe the major outcome of this study and innovative approaches.

Author Contributions:: Add details here

Author Response

  1. Introduction: Incomplete, need to elaborate more about previous studies, major scientific gaps and novelty of this study.

Response 1.

  • The introduction was expanded with more recent studies.
  • Although, there are reports applying different electric fields to bone, there are scarce studies in pure collagen and as far as we know there are not reports with the methodology we applied using prepolarization and electrophoresis method for hydroxyapatite deposition on collagen.

  1. Materials and Methods: The source, manufacturer's details are missing throughout. 

2.2. Preparation of collagen membranes. The preparation of thin collagen gels:

  • Response 2.2 -1. This was corrected, we prepared membranes

Need to prove that the Electric field on the Biomineralization of Collagen did not change the molecular pattern of collagen, which directly affect the behaviour and biological performance. Through, the authors performed FTIR and Raman spectra, its better to confirm the changes in secondary structure, amino acids composition and biocompatibility of collagen using invitro studies. 

Response 2.2-2.   If there were important changes in the molecular structure, FTIR and Raman spectroscopy are sensible enough to detect any important changes. Although we agree that conformational changes could be present, but we do not have the techniques to be able to detect those conformational changes.  On the other hand, we could not perform in vitro experiments, this will be a matter for further future studies.

Results: Unit y-Axis label is missing in all Figures, revise

This was corrected

The heading "Discussion" is missing.

We add Results and Discussion Heading

  1. Conclusions: Incomplete, describe the major outcome of this study and innovative approaches.

Response 4 The conclusions were rewritten and completed

Author Contributions:: Add details here.

Details of author contributions were added

Round 2

Reviewer 2 Report

No more comments, the authors responded well. 

Author Response

Thanks very much to the reviewers for the helpful comments.

The english has been reviewed and minor errors have been corrected through the text.
